# Intestinal-epithelial LSD1 controls goblet cell maturation and effector responses required for gut immunity to bacterial and helminth infection

Naveen Parmar[1], Kyle Burrows[2,3], Pia M. Vornewald[1], Håvard T. Lindholm[1], Rosalie T. Zwiggelaar[1], Alberto Díez-Sánchez[1], Mara Martín-Alonso[1], Madeleine Fosslie[4], Bruce A. Vallance[5], John Arne Dahl[4], Colby Zaph[6], Menno J. Oudhoff[1]*

1 CEMIR–Centre of Molecular Inflammation Research, Department of Clinical and Molecular Medicine, Norwegian University of Science and Technology, Trondheim, Norway, 2 The Biomedical Research Centre, University of British Columbia, Vancouver, Canada, 3 Department of Immunology, University of Toronto, Toronto, Canada, 4 Department of Microbiology, Oslo University Hospital, Rikshospitalet, Oslo, Norway, 5 Department of Pediatrics, Division of Gastroenterology, BC Children's Hospital Research Institute, Vancouver, British Columbia, 6 Infection and Immunity Program, Monash Biomedicine Discovery Institute and Department of Biochemistry and Molecular Biology, Monash University, Clayton, Victoria, Australia

* menno.oudhoff@ntnu.no

**Data Availability Statement:** All relevant data are within the manuscript and its Supporting Information files, except for the ATAC sequencing, which is deposited: E-MTAB-8718.

## Abstract

Infectious and inflammatory diseases in the intestine remain a serious threat for patients world-wide. Reprogramming of the intestinal epithelium towards a protective effector state is important to manage inflammation and immunity and can be therapeutically targeted. The role of epigenetic regulatory enzymes within these processes is not yet defined. Here, we use a mouse model that has an intestinal-epithelial specific deletion of the histone demethylase *Lsd1* (cKO mice), which maintains the epithelium in a fixed reparative state. Challenge of cKO mice with bacteria-induced colitis or a helminth infection model both resulted in increased pathogenesis. Mechanistically, we discovered that LSD1 is important for goblet cell maturation and goblet-cell effector molecules such as RELMß. We propose that this may be in part mediated by directly controlling genes that facilitate cytoskeletal organization, which is important in goblet cell biology. This study therefore identifies intestinal-epithelial epigenetic regulation by LSD1 as a critical element in host protection from infection.

## Author summary

The epithelium that lines our intestine has the important task of taking up nutrients, while also providing a barrier against pathogens. The intestinal epithelium performs these different tasks by having specialized cell types; enterocytes take up nutrients whereas goblet cells are in charge of producing a mucus layer. In addition, goblet cells can be stimulated to make special antimicrobial proteins. This occurs in response to cues called cytokines that come from immune cells, which are able to detect and act on the presence of pathogens such as bacteria or parasitic worms. In this study, we found that LSD1, an

**Funding:** Funding of this work was provided by the Norwegian Research Council (Centre of Excellence grant 223255/F50, and 'Young Research Talent' 274760 to MJO) and the Norwegian Cancer Society (182767 to MJO). MMA is the recipient of a Marie Skłodowska-Curie IF (DLV-794391). This work was also supported by the South-Eastern Norway Regional Health Authority, Early Career Grant 2016058, and the Research Council of Norway "Young Research Talent" grant to JAD. KB is a Banting Fellow. MF is supported by the Norwegian Research Council (grant no. 275286). CZ is supported by the Australian National Health and Medical Research Council (APP1104433 and APP1104466). The funders had no role in study design, data collection and analysis, decision to publish, or preparation of the manuscript.

**Competing interests:** The authors have declared that no competing interests exist.

enzyme that controls gene expression, was important for goblet cells. Mice that lacked LSD1 specifically in their intestinal epithelium were unable to respond to cytokines and could not defend themselves against bacterial and parasitic infections. In part, we also made use of a specific inhibitor against the enzyme activity of LSD1. This inhibitor also blocked goblet cell differentiation and goblet-cell specific antimicrobial responses to cytokines. We are thus able to manipulate epithelial responses, which may be an important tool in the future to treat patients with infectious diseases.

## Introduction

Gastrointestinal infections remain one of the most common causes for hospitalization worldwide [1]. These infections can be caused by diverse pathogens including bacteria, viruses, and parasites. Despite recent advances in treatment for infections, the increase in incidence of antibiotic resistant bacterial strains likely will ensure that infection remains a large threat to global health [2]. Therefore, increasing our understanding of disease mechanisms and exploring new options such as host-directed therapeutics remains a very important focus in basic research.

The intestinal epithelium is a crucial component of the gut barrier and is composed of a single layer of intestinal epithelial cells (IECs) including enterocytes, Paneth cells, goblet cells and tuft cells [3]. Several of these cell types are involved in the protection against pathogens. For example, Paneth cells that reside at the bottom of small intestinal crypts are a source of antimicrobial proteins such as lysozyme and α-defensins [4]. Likewise, goblet cells secrete mucins and antimicrobials that are effective against bacteria and protozoa [5]. On the other hand, tuft cells express IL-25 that is important in mounting a type 2 immune response to protect against certain helminths [6]. Finally, different pathogens elicit different immune responses that result in an alteration of the intestinal epithelial cell composition that is important for effector responses [7,8]. These epithelial effector responses are ultimately critical for immunity against intestinal pathogens.

In addition to epithelial effector mechanisms targeting pathogens, there is also a damage response upon inflammation or infection. There is a sequence of events that include migration and proliferation of epithelial cells, which eventually results in the re-establishment of mucosal homeostasis. To regain epithelial integrity after an insult, IECs are reprogrammed into a temporal reparative state that is required for repair [9,10]. Mice that lack the ability to reprogram towards such reparative state succumb from an otherwise manageable colitic insult [9,11]. However, it is unknown whether a reparative epithelial state prior to infection is beneficial for the host.

Chromatin accessibility has an important role in regulating gene expression. Chromatin accessibility can be mediated by posttranslational modifications of DNA and Histone proteins, which is done by epigenetic modifiers [12]. In the intestinal epithelium, the importance of maintaining epigenetic marks is exemplified by the finding that loss of the polycomb repressor complex 2 (PRC2) leads to a rapid loss of various cell types including intestinal stem cells, which subsequently leads to loss of tissue integrity and mice become moribund [13–15]. We recently found an important role for the demethylase LSD1 in early-life intestinal epithelial development, including the differentiation of Paneth cells [16]. LSD1 can demethylate mono or di-methylated Histone 3 Lysine 4 (H3K4me1/2) that normally mark active or poised enhancers [17]. Importantly, we previously found that LSD1-deficient small intestinal epithelium is in a continuous state of repair[16].

## Results

### LSD1-deficient epithelium is in a reparative state during homeostasis

We have recently shown that inhibition or deletion of the histone demethylase LSD1 renders the small intestinal epithelium in a continuous state of repair, and this state is beneficial for recovery after irradiation injury[16]. We generated mice that lack *Lsd1* in the intestinal epithelium specifically by crossing *Villin*-Cre mice with *Lsd1*^f/f mice (conditional knock-out (cKO) mice) and found that these mice have a near complete deletion of *Lsd1* in small intestinal epithelium (~10–20% of the epithelium still expressed LSD1 [16]). In the current study, we focus on the colon, and here we find 20–40% still expressing LSD1 in cKO mice (S1 Fig). So, the recombination in these mice is higher in the small intestine than in the large intestine. Nevertheless, we assessed whether the colonic epithelium in the cKO mice is in a general reparative state by examining different parameters such as proliferation and presence of the reparative marker Sca-1. Although we do not observe a significant increase in crypt length, we did find increased proliferation in colonic crypts of cKO mice compared to *Lsd1*^f/f (WT) littermates as measured by Ki67⁺ cell counts (Fig 1A and 1B). In support, in small intestinal crypt derived organoids, we found that the Ki67⁺ proliferative zone was expanded and with increased intensity in cKO organoids compared to WT organoids (Fig 1C and 1D). Thus, increased proliferation is an intrinsic effect of *Lsd1* deletion in intestinal epithelium. Two studies identified Sca-1 as a bona fide marker of repairing epithelium, and Sca-1-derived organoids partially retain this reparative state [9,18]. To test whether *Lsd1*-deficient epithelium contains Sca-1⁺ cells, we isolated colon crypts and analyzed Sca-1⁺ cells by flow cytometry. Indeed, we observed that crypt cells derived from cKO mice contain more Sca-1⁺ cells than their WT cage littermates (Figs 1E, 1F and S2A). Finally, IL-22 is known to be a crucial cytokine in the defense against pathogens, which among other responses is due to inducing epithelial repair [19], and by inducing antimicrobials such as REG3B and REG3γ [20]. To test if inhibition of LSD1 would affect the IL-22-induced antimicrobial responses, we co-treated WT organoids with GSK-LSD1, a potent LSD1 inhibitor that phenocopies *Lsd1* deletion [16], and IL-22. We found that IL-22 induced STAT3 Tyr705 phosphorylation by western blot, and robustly induced *Reg3b* and *Reg3g* by qPCR independently of LSD1 enzymatic activity (Fig 1G, 1H and 1I). Together, these findings suggest that the intestinal epithelium of cKO mice is in a continuous reparative state and that this reparative state does not hamper IL-22-induced antimicrobial programs.

### LSD1 is required for defense against *Citrobacter rodentium*

*Citrobacter rodentium*, a gram-negative mucosal bacterium, is used to model the pathogenesis of human enteric pathogens such as enteropathogenic *Escherichia coli* [21]. It is currently unknown whether having an epithelial reparative state prior to infection is beneficial for the host. Therefore, we orally infected cKO and WT littermates with *C. rodentium*. Strikingly, we found that cKO mice are susceptible to *C. rodentium* with increased mortality and increased weight loss compared to WT littermates (Fig 2A and 2B). We also found an increase in the number of *C. rodentium* bacteria in the stool of cKO mice compared to WT mice at day 6 (Fig 2C), but no increased *C. rodentium* numbers in liver and spleen (Fig 2D). An appropriate immune response is important in the defense against *C. rodentium*. We found near identical responses when assessing cytokine levels of IL-22, IL-17A, and IFN-γ in the colon by qPCR or in re-stimulated mesenteric lymph node cells by ELISA (Fig 2E and 2F). The intestinal barrier plays an important role in blocking the entry of pathogenic microorganisms from the lumen into the circulation. The increased susceptibility of cKO mice towards *C. rodentium* prompted us to measure intestinal permeability in animals by orally administering FITC dextran. We

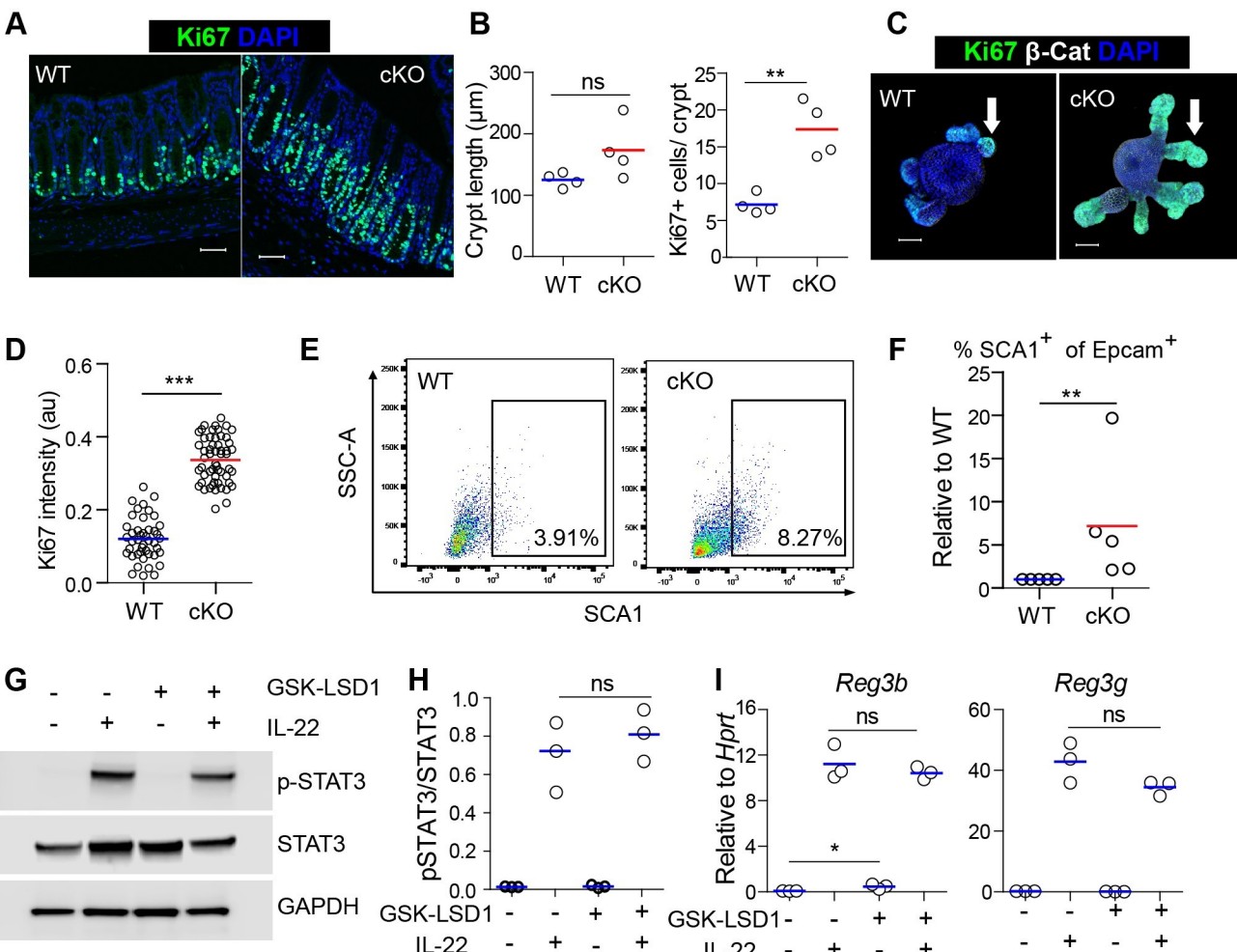

**Fig 1. Intestinal epithelial *Lsd1* deletion induces proliferation and a repair-like state in epithelium of mice.** (A) Immunofluorescent images of Ki67+ (green) in colonic epithelium of WT and cKO adult mice. DAPI was used as a nuclear counterstain. (B) Quantification of crypt length in naïve WT and cKO adult mice. Quantification of Ki67+ cells in colonic crypts of WT and cKO adult mice. n = 4 mice in each group, line is mean. (C) Organoids from WT and cKO mice were cultured for 3 days. The expression of Ki67+ (green) was determined by immunofluorescence. (D) Quantification of Ki67 intensity per organoid area in budding organoids from WT and cKO mice. Each dot represents 1 organoid. (E) Crypts cells were isolated from colon of WT and cKO adult mice and flow cytometric analysis of SCA1 was carried out. (F) Quantification of SCA1+ positive cells in crypts from WT and cKO adult mice, lines indicate littermate controls. n = 5 mice (G) Organoids were treated with GSK-LSD1 for 4 days followed by IL-22 stimulation for 30 minutes. Whole cell lysates were prepared and subjected to immunoblot analysis of p-STAT3, STAT3, and GAPDH. (H) Band intensity of pSTAT3 and STAT3 was measured using Image Studio Lite by LICOR. Each dot represents one biological replicate. (I) Organoids were co-cultured with GSK-LSD1 (5μM) and IL-22 (10 ng/ml) for 4 days in ENR (EGF, NOGGIN, R-SPONDIN) medium. The mRNA expression of *Reg3b* and *Reg3g* was determined by RT-qPCR relative to housekeeping gene *Hprt*. n = 3 biological replicates of a representative experiment. Unpaired two-tailed Student's t test (B, D,H & I) was performed to observe significant differences. ns = not significant, * P ≤ 0.05, ** P < 0.01, *** P ≤ 0.001.

observed that infection clearly led to increased permeability as measured by FITC dextran levels in the serum (Fig 2G). However, we did not observe any significant differences between infected cKO and WT mice (Fig 2G). Thus, although cKO were highly susceptible to infection, their immune response and their initial barrier function was not atypical.

## LSD1 regulates crypt length and goblet cell responses after *C. rodentium* infection

Next, we examined the epithelial response upon *C. rodentium* infection more closely. We found no differences in epithelial-derived antimicrobials, as measured by *Reg3b* and *Reg3g*

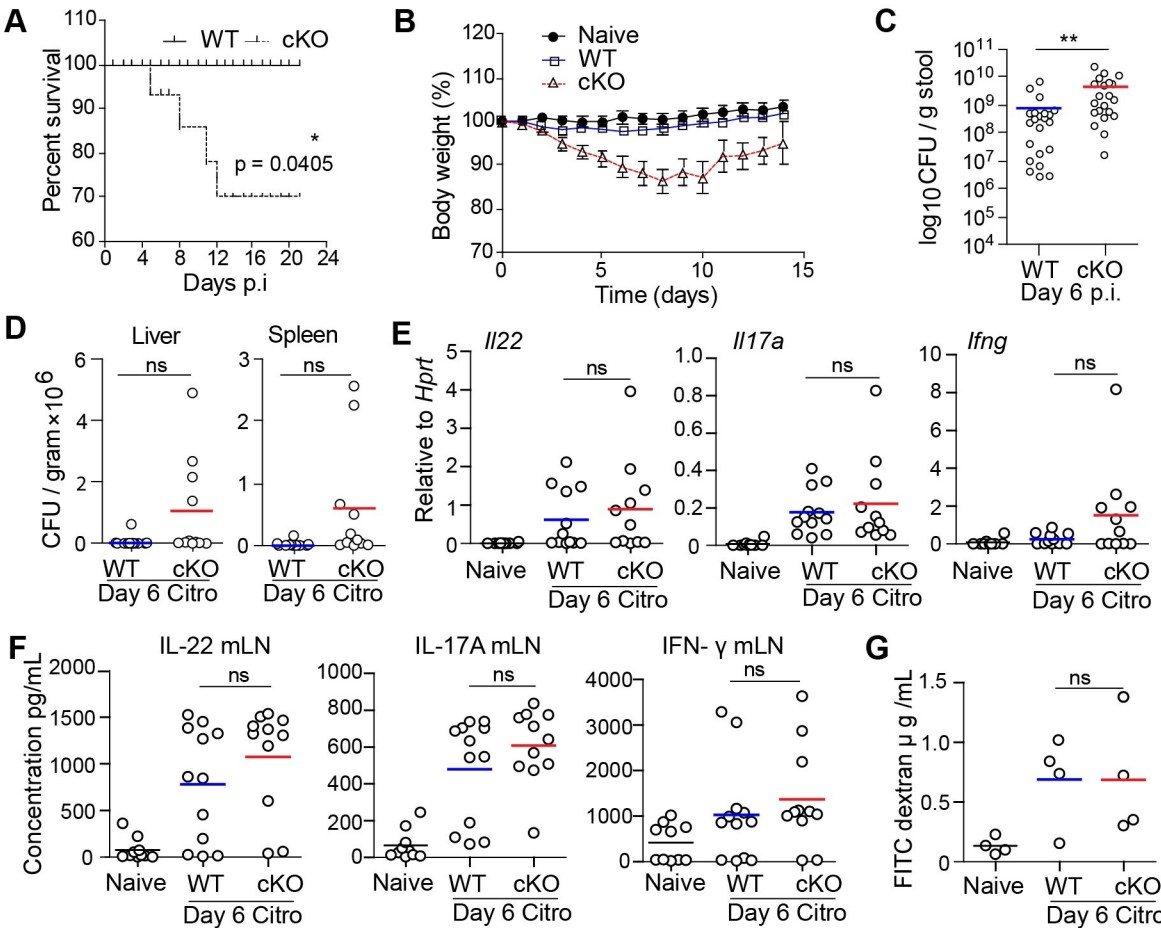

**Fig 2. LSD1-deficient epithelium renders mice susceptible to *C. rodentium*.** (A) Survival plot of WT and cKO mice after *C. rodentium* infection. Kaplan-Meier survival curves was used and tested with a Gehan-Breslow-Wilcoxon test to identify significant differences between wild-type and cKO mice (p value = 0.0405). (B) Body weight loss comparing Naive, WT and cKO mice during *C. rodentium* infection. n = 9 (naive), n = 16 (WT) and n = 14 (cKO) mice, three pooled experiments. Data points are mean and SD. (C) Stool CFU (colony forming units) counts in WT (n = 22) and cKO (n = 21) infected mice using plating on agar after 6 days of *C. rodentium* infection. (D) CFU counts in liver and spleen tissue of WT (n = 12) and cKO (n = 11) mice at day 6 post *C. rodentium* infection. (E) RT-qPCR for *Il22*, *Il17a* and *Ifng* was carried out in colon tissue from naive (n = 10), WT (n = 12) and cKO (n = 11) mice infected with *C. rodentium* for 6 days. Expression is relative to *Hprt*. (F) ELISA for IL-22, IL-17A and IFN-γ in re-stimulated mesenteric lymph node cells after 6 days of *C. rodentium* infection. (G) Measurement of FITC dextran fluoroscence in serum derived from (naive = 4), WT (n = 4) and cKO (n = 4) infected with *C. rodentium* for 6 days. * P ≤ 0.05, ** P ≤ 0.01, Each dot represents one mouse, and means are depicted.

mRNA levels in the colon comparing cKO with WT mice at day 6 post infection (Fig 3A). This indicates that an appropriate immune response and an adequate induction of epithelial-derived antimicrobials in cKO mice is insufficient to prevent morbidity and mortality. It is known that crypt length and induction of goblet cells is essential for resistance against *C. rodentium* and that these are indicators of disease [22,23]. Histological studies revealed that there is an increase in colonic crypt hyperplasia in cKO mice compared to WT mice at day 6 post infection (Fig 3B and 3C), and that these long crypts are characterized by Ki67⁺ cells (Fig 3D). Next, we sought to determine the goblet-cell responses that are induced by *C. rodentium* infection. We found that *C. rodentium* infection led to induction of the goblet-cell effector gene *Retnlb* in WT mice, however, in stark contrast, this induction was completely absent in cKO mice (Fig 3E). This was supported by RELMß staining of colon sections, where, in

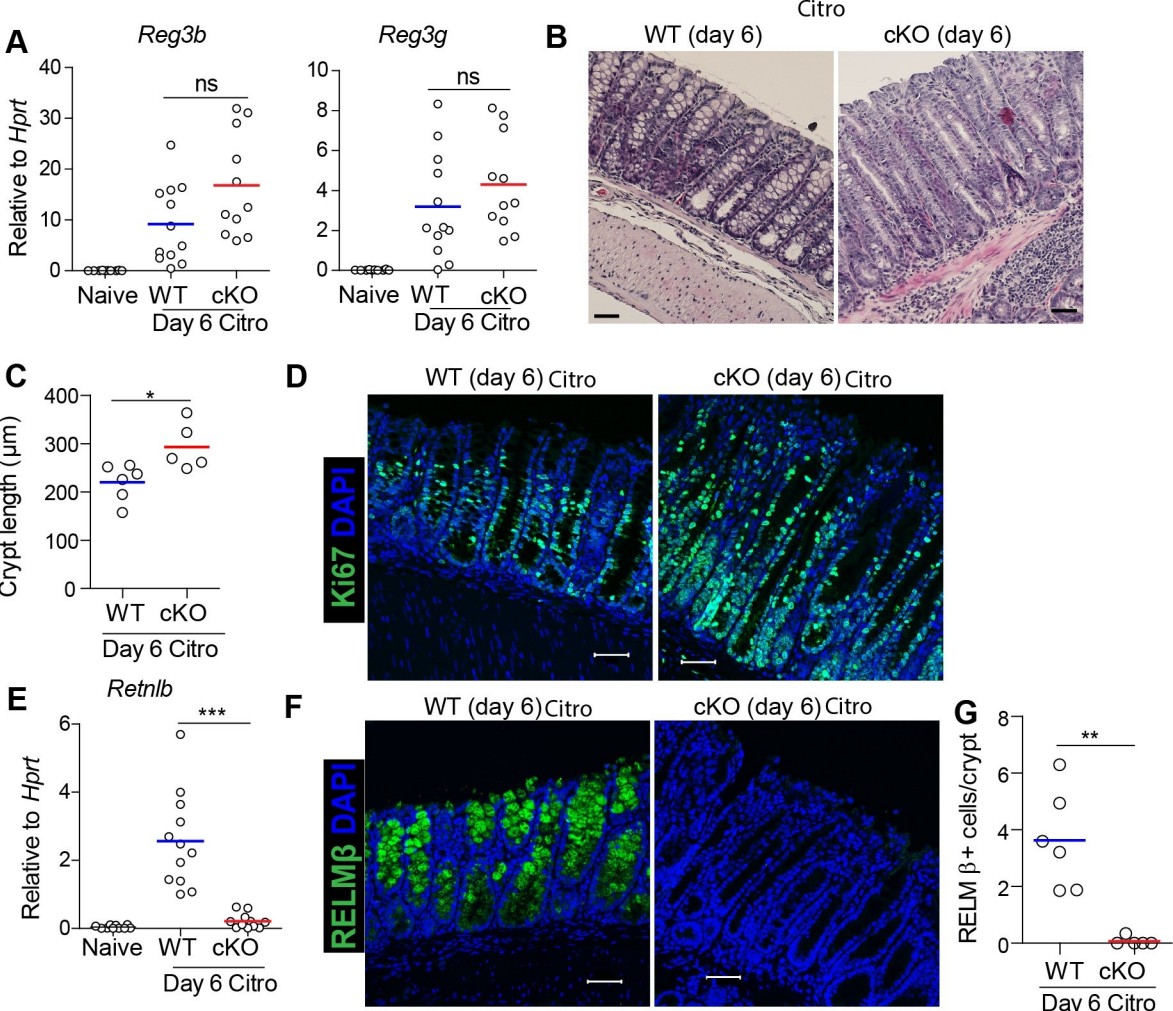

**Fig 3. Intestinal-epithelial LSD1 is required for appropriate crypt hyperplasia and goblet cell responses to infection with *C. rodentium*.** (A) RT-qPCR for antimicrobials *Reg3b* and *Reg3g* in colon tissue of Naive (WT n = 10), and WT (n = 12) and cKO (n = 11) mice infected with *C. rodentium* for 6 days. (B&C) H&E staining and quantification of crypt length at distal colon in WT (n = 6) and cKO (n = 5) mice infected with *C. rodentium* for 6 days (scale bar = 50 μm). (D) Ki67 staining to measure proliferation in distal colon tissue of WT and cKO mice infected with *C. rodentium* for 6 days. (E) RT-qPCR for goblet-cell specific antimicrobials *Retnlb* in colon tissue of Naive (WT n = 10), WT (n = 12) and cKO (n = 11) mice infected with *C. rodentium* for 6 days. (F&G) Representative confocal images and quantification of RELMβ+ cells (green) in distal colon tissue of WT (n = 6) and cKO (n = 5) infected mice. DAPI (blue) was used as a nuclear counter stain. Scale bar = 50 μm. Unpaired two-tailed Student's t test (A, C, E & G) was performed to observe significant differences among experimental groups. ns = not significant, * P ≤ 0.05, **, P < 0.01, *** P ≤ 0.001. Each dot represents one mouse and means are depicted.

infected WT mice high levels were detected, but this was virtually absent in tissues from infected cKO mice (Fig 3F and 3G). In addition, we found LSD1+ escaper crypts to be filled with RELMß+ goblet cells in cKO mice (S3A Fig). This further supports that there is an appropriate immune response in cKO mice, but cKO epithelium fails to respond properly. In summary, although IL-22-induced antimicrobials such as REG3γ are normally regulated, the proliferative and the goblet-cell responses are strongly dependent on epithelial LSD1 expression.

## LSD1 is required for optimal immunity against the helminth *Trichuris muris*

Resistance to intestinal infection is often an interplay between various cell types in which the epithelium is part of the effector mechanism. *Trichuris muris* is a murine whipworm that is closely related to the human parasite *T. trichiura*. Clearance of *T. muris* is dependent on a type 2 response that leads to increased epithelial turnover, in which the epithelium acts like an escalator to get rid of worms [24,25]. In addition, a type 2 response leads to goblet cell hyperplasia, which is also important [26]. To study the role of intestinal-epithelial LSD1 in response to *T. muris*, mice were infected with ~200 infective embryonated eggs by oral gavage and we included *Rag1*$^{-/-}$ mice as a positive control for the infection. We observed that parasite elimination was incomplete in cKO mice on day 21 post infection with similar worm counts as susceptible *Rag1*$^{-/-}$ mice, at which point nearly all WT littermates had cleared the infection (Fig 4A). Goblet cell hyperplasia is a hallmark feature and important for the elimination of worms from the ceacum [26,27]. Therefore, we carried out a histological examination of cKO and WT ceacums at day 21 post *T. muris* infection. Our Periodic acid-Schiff (PAS) staining revealed a near complete absence in goblet cells in the ceacum of *T. muris* infected cKO mice compared to infected WT mice (Fig 4B and 4C). In support, we found reduced expression of goblet-cell effector genes *Retnlb* and *Clca1* in infected cKO mice compared to WT mice (Fig 4D) as well as a stark reduction in RELMß$^+$ cells per crypt (Fig 4E and 4F). Similar to the response to *C. rodentium*, we found increased crypt length in cKO mice compared to WT mice (Fig 4G). Elimination of *T. muris* parasites is dependent on expression of type 2 cytokines. Analysis of the hallmark Th2 cytokine revealed that protein expression of IL-13 was almost similar in WT and cKO mice infected with *T. muris* (Fig 4H). However, the expression of the Th1 cytokine IFN-γ was upregulated in restimulated lymph node cells of cKO mice in comparison with WT littermates (Fig 4H). Therefore, we conclude that upregulated IFN-γ expression in infected cKO mice in combination with a lack of goblet cell responses renders these mice susceptible to *T. muris* infection.

## LSD1 controls homeostatic and cytokine driven goblet cell responses

In both *C. rodentium* and *T. muris* infection models we found a complete inability to induce goblet-cell effector molecules that are common in response to these infections. We have previously found a modest reduction in goblet cell numbers in small intestinal tissues of cKO mice [16]. However, we found that goblet cell maturation is much more impaired in the colons of naïve cKO mice, as shown by reduced numbers and by a lack of large MUC2$^+$ and Ulex Europaeus Agglutinin-1 (UEA1)$^+$ granules (Fig 5A and 5B). Similarly, we also found significant reduction of MUC2 protein levels and *Muc2* gene expression in colon organoids derived from cKO mice compared to those from WT mice (Figs 5C, 5D, S5A and S5B). In support, we find that GSK-LSD1 also drastically inhibits goblet cell maturation in small intestinal derived organoids (S4A and S4B Fig).

To test if we could rescue the lack of goblet cell maturation in LSD1-deficient intestinal epithelium, we treated organoids with the γ-secretase inhibitor DAPT, a well-established NOTCH inhibitor that leads to induction of goblet cells [28]. In addition, to define cytokine-induced goblet cell responses, we treated WT and cKO organoids with IL-13 or IL-22. As a readout, we used immunofluorescent staining of MUC2, a canonical goblet cell marker, and RELMβ, a goblet cell effector protein, both of which were normalized to organoid area. Expectedly, DAPT treatment induced goblet cell differentiation in WT organoids while limiting organoid growth (Fig 5E–5J). In contrast, although DAPT treatment led to reduced growth in cKO organoids, it did not significantly induce MUC2 levels per organoid area (Fig 5E and 5F).

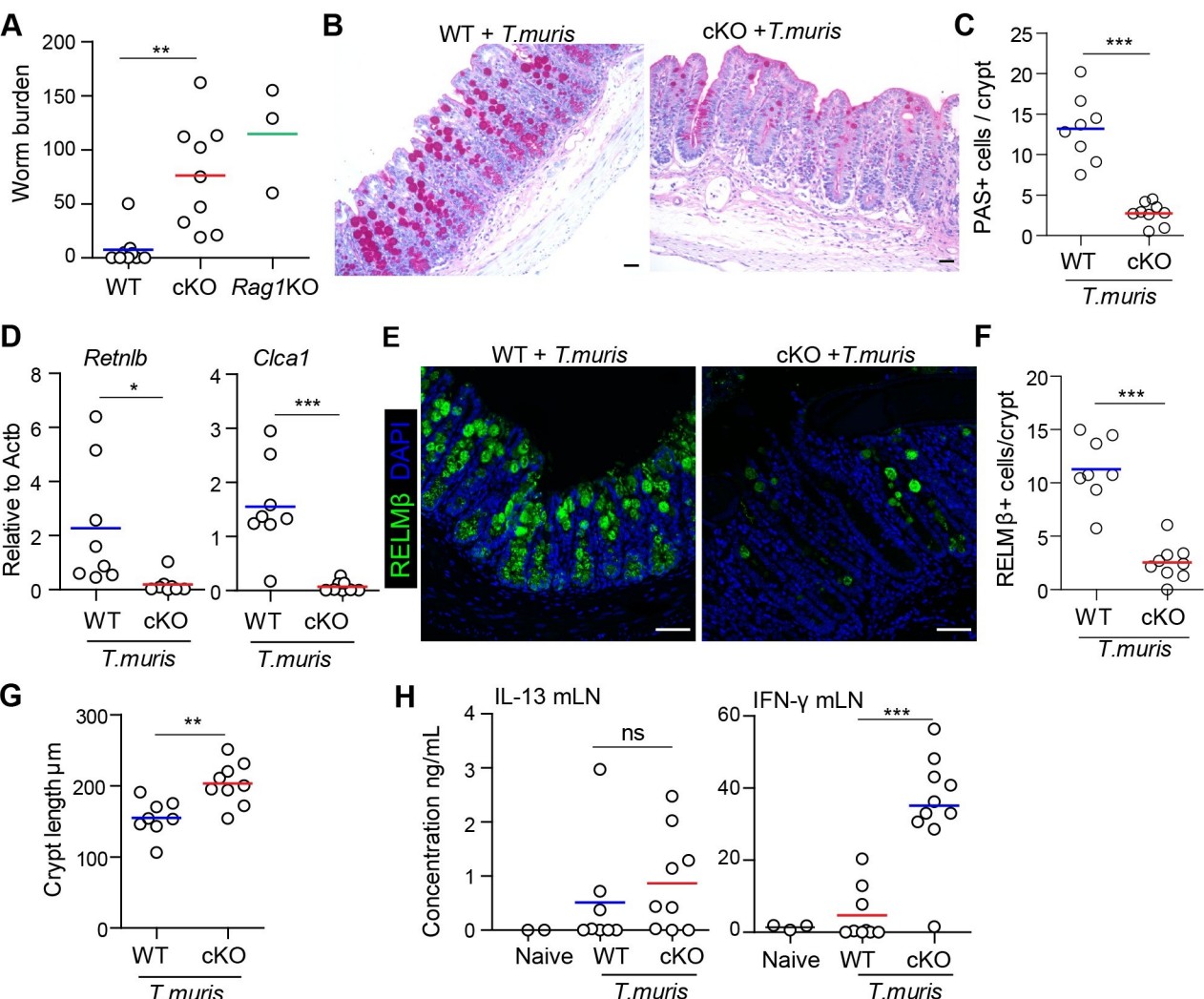

**Fig 4. Intestinal-epithelial LSD1 is required for effective immunity against the whipworm *Trichuris muris*.** (A) WT (n = 9), cKO (n = 9) and *Rag1*KO (n = 3) mice were infected with 200 embryonated eggs of *T. muris*. Worm burden was quantified in caecum after 21 days of infection. Pooled from 2 independent experiments. (B) PAS staining for goblet cells was done in ceacal epithelium of WT and cKO mice after 21 days of *T. muris* infection. (C) Quantification of PAS+ cells in caecal epithelium of WT (n = 8) and cKO (n = 9) mice after 21 days of *T. muris* infection. (D) RT-qPCR for *Retnlb* and *Clca1* in proximal colon of WT (n = 8) and cKO (n = 8) after 21 days of *T. muris* infection. (E&F) Representative confocal images and quantification of RELMβ+ cells (green) in caecum tissue of WT (n = 8) and cKO (n = 9) mice infected with *T. muris* for 21 days. DAPI (blue) was used as a nuclear counter stain. Scale bar = 50 μm. (G) Measurement of cecal crypts length in WT (n = 8) and cKO (n = 9) mice infected with *T. muris* for 21 days. (H) Sandwich ELISA for quantification of IL-13 and IFN-γ expression in restimulated mesenteric lymph node cells after 21 days of *T. muris* infection. Unpaired two-tailed Student's t test (A, C, D, F&G) and one-way ANOVA (H) was performed to observe significant differences. ns = not significant, * P ≤ 0.05, **, P < 0.01, *** P ≤ 0.001.

Similar to recent findings, IL-13 but not IL-22 robustly induced MUC2 in WT organoids [8], and this response was similar in cKO organoids (Fig 5E and 5F), suggesting that, in contrast to DAPT treatment, IL-13 can overrule the lack of goblet cell differentiation caused by LSD1 deficiency. DAPT did not induce RELMβ in either WT or cKO cultures (Fig 5G and 5H), which suggests NOTCH inhibition is not sufficient for RELMβ effector responses. In contrast, both IL-22 and IL-13 caused a dramatic increase in RELMβ+ cells in WT organoids, whereas there was only a subtle response in cKO organoid cultures (Fig 5G and 5H). Together, only IL-13 is able to stimulate MUC2+ goblet cell numbers in cKO organoids, but this is insufficient to induce the effector RELMβ. As IL-22 is a stronger inducer of RELMβ in WT organoids, we

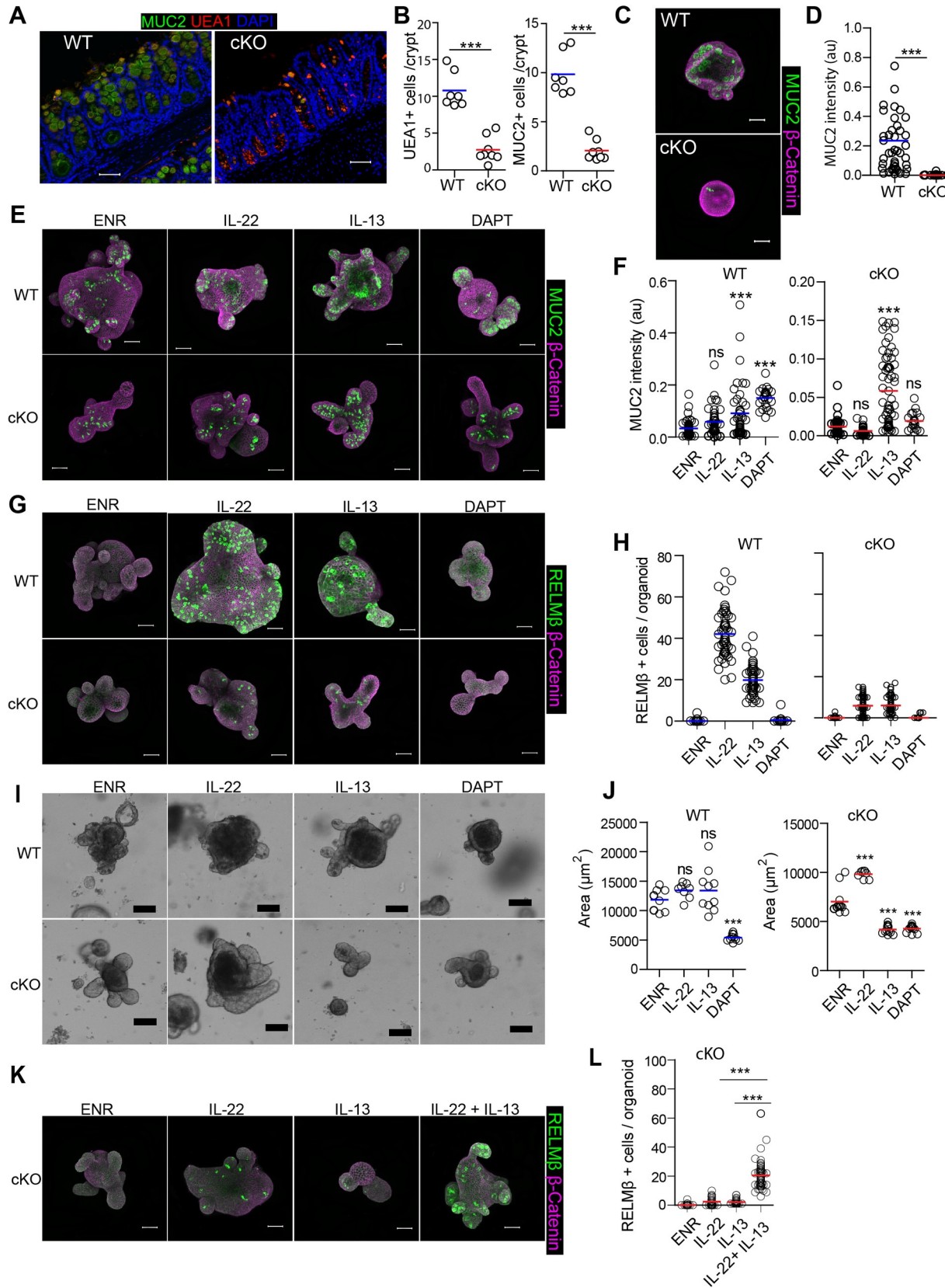

**Fig 5. LSD1 controls (immune-driven) goblet cell responses.** (A) Colon sections of WT and cKO mice were stained for MUC2 (green), Ulex europaeus agglutinin I (UEA1) (red). UEA1 is a lectin which binds to glycoproteins and glycolipids and used as an additional marker for goblet cells. DAPI (blue) used as a counterstaining. (B) Quantification of MUC2+ cells and UEA1+ cells in the colon of WT (n = 7) and cKO (n = 8) mice. (C) WT and cKO colon organoids were seeded in ENR for 4 days and immunofluorescent staining of MUC2 (green) and β- catenin (purple) was performed. (D) Quantification of MUC2 intensity per colon organoid is represented. Colon organoids were derived from WT (n = 2) and cKO (n = 2) mice. (E&G) Immunofluorescent confocal images for MUC2 (E, green) and RELMβ (G, green) in WT and cKO organoids left untreated in ENR or supplemented with IL-13 (10 ng/ml), IL-22 (5ng/ml) or DAPT 10 μM for 4 days. DAPI was used to stain nuclei (blue). (F&H), Quantification of MUC2 fluorescence intensity (F) and number of RELMβ positive cells (H) per organoid. (I) Representative bright field images of WT and cKO organoids treated with IL-13 (10 ng/ml), IL-22 (5ng/ml) and DAPT 10 μM for 4 days. Scale bar is 100μm. (J) Quantification of organoid area from minimal projections 4 days after seeding. Dots represents the mean of one well. Images are representative of 3 independent experiments. (K) cKO organoids were treated with IL-13 (10 ng/ml), IL-22 (5ng/ml) and combination of IL-13 + IL-22 for 4 days. Immunofluorescent images of RELMβ (green) and β-catenin (purple). (L) Quantification of RELMβ+ cells per organoid are shown. Unpaired two-tailed Student's t test (B & D) and One-way analysis of variance with Tukey's Multiple Comparison Test (F, H, J& L) was performed to observe significant differences. ns = not significant, $^*$ P $\leq$ 0.05, $^{**}$, P < 0.01, $^{***}$ P $\leq$ 0.001.

reasoned that perhaps incubation of IL-13, to allow for goblet cell differentiation, and IL-22 would induce RELMβ. Indeed, cKO organoids treated with both IL-13 and IL-22 led to a distinct induction of RELMβ$^+$ cells (Fig 5K and 5L).

## LSD1 inhibition results in rapid reduction of goblet cell maturation

Next, we resorted to a previously performed RNA seq experiment comparing small intestinal crypt cells from cKO mice with WT littermates, as well as RNA seq from 24h GSK-LSD1 treated organoids compared to untreated organoids [16]. Although these datasets were not generated from colon epithelium, we anticipated that LSD1-controlled mechanisms would be sufficiently comparable between colon and small-intestinal epithelium to provide mechanistic insight. Predictably, using gene set enrichment analysis, we found a negative correlation with a goblet cell gene set [29], both in the data from *in vivo* crypts and *in vitro* organoid experiment (Fig 6A). This suggests that within 24h, thus almost immediately after LSD1 inhibition, there is a block in goblet cell maturation.

## LSD1 inhibits genes associated with cytoskeletal organization by controlling chromatin accessibility

To identify other prominent gene programs or processes regulated by LSD1, we performed an unbiased analysis of gene ontology (GO) terms associated with the top 500 upregulated genes in cKO crypt cells compared to WT crypt cells (Fig 6B). We found several terms associated with the regulation of actin and cytoskeletal processes to be enriched in cKO crypts (Fig 6B). Importantly, unlike the goblet cell gene set signature, the genes associated with 'cytoskeletal organization' GO terms that are higher expressed in cKO crypts were not affected upon 24h GSK-LSD1 treatment in organoids (Fig 6C). We previously found that genes upregulated in cKO crypt cells associated with increased H3K4me1 levels at putative enhancer sites [16]. To test if increased H3K4me1 levels also led to functional changes in chromatin accessibility, we performed Assay for Transposase-Accessible Chromatin using sequencing (ATAC-seq) comparing WT and cKO small intestinal crypts. We found that genomic sites with significant H3K4me1 peaks that were identified previously also had increased ATAC levels, suggesting increased accessibility (Fig 6D). This also included genes associated with cytoskeletal organization, for example *Flna*, the gene for FILAMIN A (Fig 6E). In support, using a published LSD1 ChIP seq data set from a human cancer cell line [30], we found that LSD1 is located at the *FLNA* locus where there is also H3K4me1 (Fig 6F). Thus, this suggests that LSD1, by demethylating H3K4me1 levels at putative enhancer sites, directly controls these important mediators of the actin cytoskeleton. Knockdown of *Flna* in intestinal tumor organoids led to reduced stem cell marker expression and proliferation, suggesting it could affect proliferation and

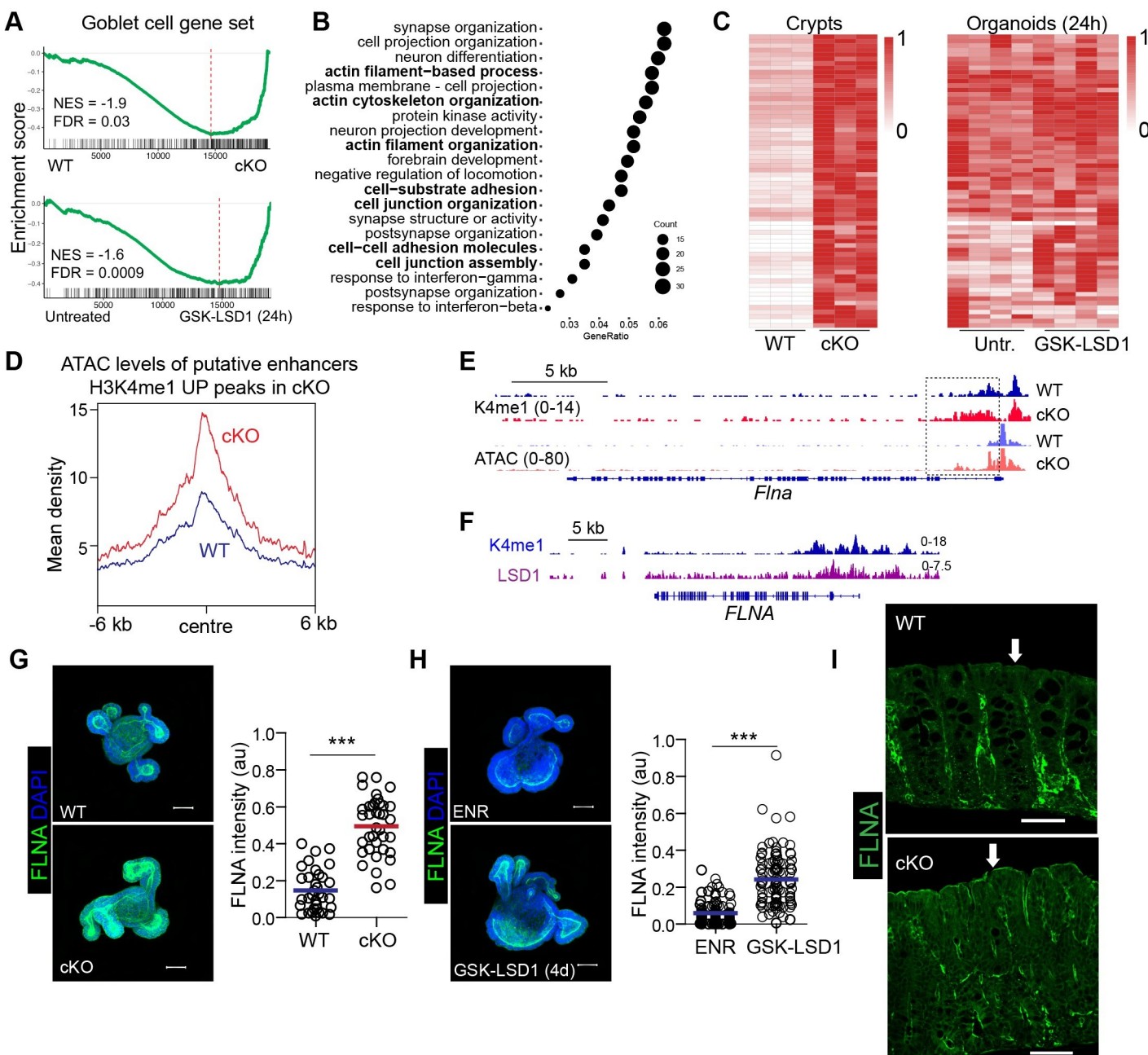

**Fig 6. LSD1 epigenetically controls modulators of the cytoskeleton.** (A) Application of GSEA of a goblet cell gene set in WT and cKO small intestine RNA seq, and untreated vs GSK-LSD1 (24h) treated organoids RNA-seq data. (B) Top 500 genes upregulated in cKO crypts were selected and unbiased association with the top 20 GO terms is displayed. (C) Genes taken from in 6B bold-selected GO terms are displayed in a heatmap using Transcripts Per Million (TPM) values scaled to the max of the gene comparing WT and cKO crypts (left) and untreated and GSK-LSD1 (24h) treated organoids (right). (D) Mean ATAC signal of normalized peak density surrounding previously identified H3K4me1 peaks that are significantly up in cKO crypts compared to WT crypts. Representative graph from n = 2. (E) Representative Integrative Genomics Viewer (IGV) tracks of H3K4me1 and ATAC signal at the *Flna* locus (n = 2). (F) IGV tracks of H3K4me1 and LSD1 levels at the *FLNA* locus in a human cancer cell line. (G) Representative confocal images of FILAMIN A (FLNA) stained in WT and cKO organoids. Quantification of FLNA intensity in WT and cKO organoids. (H) WT organoids were untreated (ENR) or treated (GSK-LSD1- 5 μM) for 4 days and then stained for FLNA (green). DAPI was used as a counter stain (blue). Scale bar = 50 μm Organoids were quantified for FLNA fluorescent intensity. (I) FLNA staining in paraffin embedded colon sections of naive WT and cKO mice. White Arrow indicates the increased FLNA expression on the apical side of colonic crypts. Scale bar = 50 μm. Unpaired Student's t test was performed to observe significant differences. *** P ≤ 0.001.

differentiation [31], and thus we explored its expression in organoids and colon of WT and cKO mice. We found increased protein level of FILAMIN A (FLNA) in cKO organoids and in organoids that were treated long-term with GSK-LSD1 (Fig 6G and 6H). Finally, increased FLNA protein levels were found *in vivo* in cKO colon epithelium where it accumulated at the apical side whereas it was virtually absent in WT colon epithelium (Figs 6I and S3B). Together, we suggest a model where changes in expression of cytoskeletal modulators such as FLNA depends on remodelling of chromatin accessibility.

## LSD1-controlled cytoskeletal organization partially control goblet cell responses

We found that LSD1 is required for goblet cell maturation and effector responses, and that direct NOTCH inhibition by DAPT treatment was insufficient to overcome this deficiency in cKO epithelium (Fig 5E). However, IL-13, which also induces goblet cells by inhibiting NOTCH signalling, was able to induce MUC2 expression (Fig 5E). In addition, IL-13 also affected cKO but not WT organoid growth (Fig 5I and 5J). Importantly, IL-13 does not only inhibit NOTCH, it is also known to induce cytoskeletal changes, which is perhaps best described in macrophage differentiation [32]. As the expression of cytoskeletal modulators is controlled by LSD1, we reasoned that IL-13 thus can overcome this deficiency by affecting both cytoskeletal features and NOTCH signalling. Furthermore, the cytoskeletal network itself plays an important role in intestinal epithelial cell differentiation [33]. In this study, Petersen *et al.* used the ROCK inhibitor Y-27632 in mice and organoids and found that this altered cytoskeletal properties and resulted in increased secretory cell differentiation including goblet cells. To test if ROCK inhibition would induce goblet cell maturation in cKO epithelium, we treated cKO-derived organoids with Y-27632. We found that Y-27632 treatment resulted in reduced growth, but modestly increased MUC2 levels per organoid (Fig 7A–7D). To test if altering ROCK inhibition would also allow for IL-22 mediated goblet cell effector responses, we treated cKO organoids with a combination of IL-22 and Y-27632. We found that Y-27632 limited IL-22 induced growth, and also limited organoid budding (Fig 7E and 7F). Further-more, Y-27632 enhanced RELMβ expression in IL-22 treated cKO organoids compared to IL-22 treatment alone (Fig 7G and 7H). Similarly, using sustained GSK-LSD1 treated organoids, we found that Y-27632 also induced RELMβ expression in IL-22 treated organoids (Fig 7I and 7J). Together, this work suggests that LSD1 may affect goblet cell differentiation and effector responses in part by controlling the cytoskeletal organization.

## Discussion

The gastrointestinal system is continuously exposed to various pathogens, chemicals, and die-tary components that can result in inflammation and tissue damage. It was previously found that the epithelium, after damage or infection, temporarily transitions into a reparative fetal-like state that is necessary to regain homeostasis [9,18]. Thus, drugs that would induce this transition of the epithelium could be attractive therapeutics used in patients with infection or inflammatory bowel disease to aid in the tissue repair, or even prevent disease. Indeed, in a previous report, we showed that reparative epithelium is beneficial upon irradiation injury and that LSD1 is a druggable target that could facilitate the reprogramming [16]. In addition, treat-ing mice with GSK-LSD1 upon damage alleviates colitis symptoms in DSS-colitis [34]. In this study, we tested whether a pre-existing epithelial reparative state, such as in LSD1 cKO mice, would be beneficial in infectious disease models. We found that LSD1 cKO mice are suscepti-ble to two different infectious disease models. Mechanistically, we find clear evidence that gob-let cell effector genes rely on LSD1, both under homeostasis and upon induction in the diverse

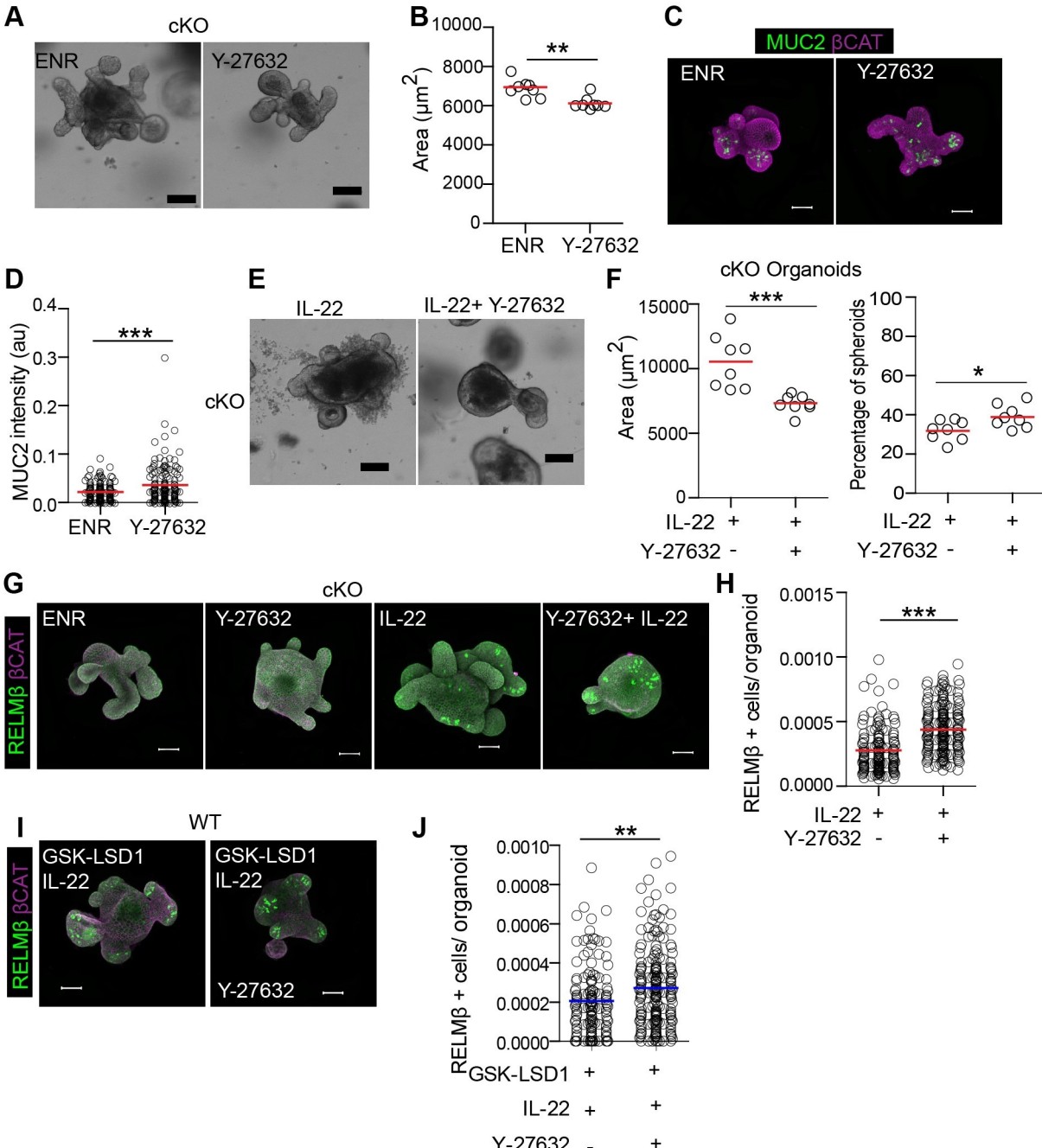

**Fig 7. Cytoskeleton manipulation partially rescues goblet cell maturation in LSD1-deficient organoids.** (A) cKO organoids were left untreated (ENR) or treated with Y-27632 (10 μM) for 4 days. Representative bright field images are shown. Scale bar is 100 μm. (B) Quantification of organoid area from minimal projections 4 days after seeding. Dots represents mean of one well, pooled from 3 independent experiments. (C&D) Representative immunofluorescent images of MUC2 (green) and β-catenin (purple). MUC2 fluorescence intensity relative to organoid area was measured. Scale bar is 50 μm. (E) cKO organoids were treated with IL-22 (5ng/ml) or IL22+ Y-27632) for 4 days. Representative bright field images are shown. Scale bar is 100 μm. (F) Mean organoid area of cKO organoids is depicted as well as the percentage of spheroids in each well 4 days after seeding. (G) cKO organoids were left untreated (ENR) or treated with IL-22 (5ng/ml), Y-27632 (10 μM) and combination of Y-27632 and IL-22 for 4 days and representative images for RELMβ (green) and β-catenin (purple)staining are shown. (H) Pooled quantification of RELMβ+ cells per organoid area from independent experiments. (I) WT organoids were pre-treated with GSK-LSD1 for 7 days. Next, these organoids were treated with combination of IL-22 (5ng/ml) + GSK-LSD1 and GSK-LSD1+ IL-22+ Y-27632 for 4 days. Immunofluorescence images of RELMβ (green) and β-catenin (purple) are shown. (J) Quantification of RELMβ+ cells per organoid area is shown. Unpaired two-tailed Student's t test (B, D, F, H & J) was performed to define significance. * P ≤ 0.05, **, P < 0.01, *** P ≤ 0.001.

disease models. In addition, using an unbiased analysis of previous RNA-seq data, we found that genes associated with modulation of the cytoskeleton are regulated by LSD1. As an example, we find that the gene encoding for FILAMIN A, a protein involved in mechanosensing and mechanotransduction [35], is likely a direct target of LSD1 demethylase activity. This could also explain how LSD1 mediates genes that are controlled by the mechanosensing regulatory proteins YAP/TAZ, including the reparative gene set, such as we found previously [16]. Further studies are needed to elucidate the exact role of FILAMIN A and other cytoskeletal modulators in LSD1-controlled intestinal epithelial biology.

LSD1 was the first histone demethylase discovered and has been primarily studied in developmental biology and cancer. Of note, it is currently a target in several clinical trials for the treatment of cancers. Our data suggests that these patients may experience higher susceptibility to gastrointestinal insults. Nevertheless, the mode of action of LSD1 in the various systems is not always clear. For example, it was found that inhibition of LSD1 leads to differentiation of myeloid leukemia cells, but that this was due to the scaffolding function of LSD1, rather than its histone demethylase activity [36]. This last part fits with our finding of the immediate loss of goblet cell differentiation upon inhibition of LSD1, which could be independent of H3K4me1/2 enhancer modulation by LSD1, which we would expect to lead to more transient changes in cell lineage choice. Indeed, LSD1 inhibition can lead to the disassociation of the LSD1-GFI1 co-repressor complex as was previously suggested and reported [16,36]. In support, GFI1 KO mice have reduced goblet cell differentiation maturation [37]. The classical mode of action by LSD1 however is demethylation of enhancer regions [38]. The kinetics that would affect these genes may be a bit slower, and this is indeed what we observe in the regulation of cytoskeletal-modulation associated genes such as *Flna*. We propose that this latter mechanism is also important in the control of goblet cell differentiation and effector responses, such as those induced by IL-13 and IL-22.

## Material and methods

### Ethics statement

Mice were maintained at the Comparative Medicine Core Facility (CoMed) at NTNU and at the Biomedical Research Centre at the University of British Columbia. *C. rodentium* experiments were ethically approved by the Norwegian Food Safety Authority (FOTS ID 11842). *T. muris* experiments were approved by the University of British Columbia Animal Care Committee (protocol number A13-0010) in accordance with the Canadian Guidelines for Animal Research.

### Mouse strains

*Villin-Cre* were a kind gift from Sylvie Robine [39], *Lsd1*^f/f mice were a kind gift from Dr. Stuart Orkin [40], and *Rag1*^-/- mice were obtained from Jackson laboratories. Mice breeding was carried out and were housed at the Comparative Medicine Core Facility (CoMed).

### *Citrobacter rodentium* infection and CFU counts

*Citrobacter rodentium* infection studies were performed following the guidelines and recommendations for the care and use of animals in research and were approved by Norwegian Food Safety Authority (FOTS ID: 11842). Briefly, GFP-expressing *C. rodentium* [22] was grown at 37°C in Luria- Bertani (LB) medium supplemented with chloramphenicol (30 μg/ml). 6–10 weeks old male and female littermates were infected by oral gavage with overnight culture of $10^8$–$10^9$ CFU per mouse delivered in a volume of 0.1ml sterile PBS. Mice were daily

monitored after the oral administration with *C. rodentium* and assessed for weight loss and other pain associated behaviors like hunchbacked posture, rectal prolapse and piloerection. Fecal samples were collected, weighed and homogenized in a sterile PBS using FastPrep homogenizer from MP biomedicals. Serially diluted homogenates were plated on chloramphenicol resistant agar plates and number of colonies were counted after incubation of plates at 37°C for 18–24 h. At the end of every experiment, mice were humanely euthanized by inhalation of $CO_2$.

### *Trichuris muris* infection in mice

*Trichuris infections* were performed at the University of British Columbia following institutional guidelines. Briefly, both male and female littermates were infected by oral gavage with high dose of approximated 200 embryonated *T. muris* eggs. After 21 days of *T. muris* infections, mice were sacrificed for the analysis of worm burden and other gut associated parameters in the ceacum.

### Immunofluorescence and immunohistochemistry staining of tissue sections

After sacrificing the mice, segments of intestinal mouse were dissected and washed with sterile PBS and then rolled in a swiss roll. Intestinal tissue was fixed with 4% formaldehyde for 48 h at room temperature. Tissue was subsequently dehydrated through a series of ethanol graded and then embedded in the paraffin wax blocks. 5 μm thick sections were prepared using a microtome and tissue was placed on glass slides. The tissue slides were then deparaffinized and rehydrated, followed by antigen retrieval (Citrate buffer pH-6.0 for all staining and Tris EDTA buffer pH-9.0 for Filamin A). Sections were blocked in blocking buffer (1% BSA and 2% normal goat serum) for 1 h at room temperature in a humidified chamber. Tissue were incubated with diluted primary antibody (LSD1, 1:200 Cell Signaling Technology 2184, MUC2 1:200 Santa Cruz Biotechnology sc-15334, anti Ki67 1:200 Invitrogen MA5-14520, RELMβ 1:1000, Peprotech 500-p215, Filamin A 1:200 Invitrogen PA5-82043 overnight at 4°C. Next day, slides were washed with 0.2% PBST for 10 min. Tissue was incubated with secondary antibody coupled to fluorochromes, UEA-1 (1:500 Vector Laboratories RL-1062) and counterstained with DAPI for 1 h at room temperature in the dark. After incubation, slides were washed 3 times with 0.2% PBST for 10 mins each and then mounted in Fluoromount G. Hematoxylin and Eosin (H&E) staining and Periodic Acid Schiff (PAS) staining in paraffin sections was performed according to the standard histological procedures.

### RNA isolation and qPCR

Total RNA was isolated from organoids and colon tissue using the Quick-RNA MicroPrep Kit (Zymo Research). Briefly, organoids were first lysed in RNA-Solv Reagent provided by Omega Biotek. For RNA isolation from colon tissue, small piece of colon tissue was first homogenized using FastPrep homogenizer and was resuspended in RNA-Solv Reagent. After lysis, cell lysates were centrifuged at $\geq$12,000 x g for 1 minute for the removal of particulate and supernatant was transferred to a fresh tube. Equal volume of ethanol (95–100%) was added to the sample and mixture was centrifuged in Zymo-Spin IC Column. DNase treatment was performed by adding DNase mix directly on to the column and incubated for 15 minutes. To wash the column, 400 μl RNA Prep Buffer to the column and centrifuge for 30 seconds and followed by 700 μl RNA Wash Buffer to the column and centrifuge for 30 seconds. To ensure complete removal of wash buffer, 400 μl RNA of Wash Buffer was added and column was centrifuged for 2 minutes at 12,000 x g. RNA was eluted into the fresh tube in a volume of 20 μL

for organoids and 40 μL for colon tissue by addition of DNase/RNase-Free water onto the column. The eluted RNA was used immediately or stored at -80˚C for downstream use. Equal amount of purified RNA was reversed transcribed to cDNA using the Applied Biosystem High-Capacity RNA-to-cDNA Kit. Quantitative real-time PCR was performed using the StepOnePlus Real-Time PCR System by using both TaqMan Universal PCR Master Mix and SYBR Green PCR Master Mix by Applied Biosystem. All the primers used are represented in the supplementary table (S1 Table).

## RNA Seq analysis

RNA sequencing data from intestinal epithelium with conditional LSD1 KO (accession E-MTAB-7862) and small intestinal organoids treated with GSK-LSD1 (accession E-MTAB-9077) were analyzed as described previously [16]. GSEA analysis was run with log2fold change as weights, 10000 permutations and otherwise default settings using the R-package clusterProfiler [41]. The clusterProfiler package was also used for GO-term analysis for terms in biological process. Heatmaps were created using the R-package pheatmap.

## ChIP Seq analysis

H3K4me1 ChIP sequencing data from intestinal epithelium with conditional LSD1 KO was analyzed as described previously [16] (accession E-MTAB-7871) and processed bigwig files from LSD1 ChIP sequencing where downloaded from accession GSE66298 [30]. Plots of ChIP seq signal where made using integrative genomics viewer from bigwig files [42].

## ATAC-seq

Sorted (Epcam$^+$) small intestinal crypt cells were harvested and frozen in culture media containing FBS and 10% DMSO. Cryopreserved cells were sent to Active Motif to perform the ATAC-seq assay. The cells were then thawed in a 37˚C water bath, pelleted, washed with cold PBS, and tagmented as previously described [43], with some modifications based on [44]. Briefly, cell pellets were resuspended in lysis buffer, pelleted, and tagmented using the enzyme and buffer provided in the Nextera Library Prep Kit (Illumina). Tagmented DNA was then purified using the MinElute PCR purification kit (Qiagen), amplified with 10 cycles of PCR, and purified using Agencourt AMPure SPRI beads (Beckman Coulter). Resulting material was quantified using the KAPA Library Quantification Kit for Illumina platforms (KAPA Biosystems), and sequenced with PE42 sequencing on the NextSeq 500 sequencer (Illumina).

ATAC-seq data was analysed with the same pipeline as analysis of ChIP-Seq data described in [16].

## Mouse duodenum crypt isolation and organoids culture

We followed standard protocol for crypts isolation as described [45]. Briefly, we isolated 10 cm duodenum section of the WT and cKO mice. The duodenum was washed with ice-cold PBS and open longitudinally. Next, villi were scraped off using a glass coverslip and washed with ice-cold PBS. The duodenum tissue was cut into 2–4 mm small pieces and transferred to 50 mL falcon tube. These fragments were washed with ice cold PBS by up and down with a 10mL pipette coated with FBS. This step was repeated 10 times until supernatant turned fully clear. Next, these fragments were incubated with 30 mL ice-cold crypt isolation buffer (2mM EDTA in PBS) with gentle rotation in the cold room. After 30 minutes, fragments were settled down and supernatant was discarded. After this, fragments were pipette up and down with 20ml of PBS to release the villus and crypts. This step was repeated until most of crypts were released

and passed through the 70-μm cell strainer and collected into FBS coated sterile 50ml falcon tube. The filtered fractions contain crypts and were spun down at $300 \times g$ for 5 min and pellet was resuspended in 10 mL of ice-cold basal culture media. This fraction was washed with ice-cold basal culture media to remove the single cell at $200 \times g$ for 2 mins. Next 10 μL of crypts from 10 mL resuspended fractions was taken and counted under light microscope. After this counting, crypts were centrifuged and resuspended in thawed Matrigel and 40 μL of the resuspended Matrigel droplet was added to well in pre-warmed 24 well plate. Organoids were grown and maintained in 'basal crypt medium' (Advanced DMEM/F12 supplemented with penicillin/streptomycin, 10 mM HEPES, 2 mM Glutamax, N2 (Thermo Fisher, 17502048), B-27 (Thermo Fisher, 17504044)) supplemented with N-acetyl-L-cysteine (Sigma, A7250), 50 ng/ml murine EGF [ThermoFisher, PMG8041], 20% R-spondin-1 conditioned medium, (kind gift from Dr. Calvin Kuo), and 10% Noggin conditioned medium (kind gift from Dr. Hans Clevers). The organoids were maintained and passaged every 5–6 days regularly in incubator with 5% $CO_2$ at 37˚C.

## Colon crypts isolation and colon organoid cultures

For isolation of colonic crypts, colon from WT and cKO mice was dissected and flushed with PBS. Next, distal colon was further taken for the isolation of colonic crypts. Colon was opened longitudinally and cut into small pieces (2–4 mm) and placed in 50 ml falcon having approximately 30 ml ice-cold PBS. These small fragments were washed with 30 ml of ice-cold PBS by pipetting up and down using 10 ml pipette coated with sterile FBS. After 5–6 washes, fragments were incubated with crypts isolation buffer (10 mM EDTA in PBS) for 45 minutes in cold room with slight agitation on rocker. After incubation, the crypt isolation buffer was discarded and 20 ml ice cold PBS was added. Next, to detach the crypts, vigorous pipetting with 10 ml pipette was performed and supernatant containing crypts was collected into fresh 50 ml tube coated with sterile FBS. This step was performed 5 to 6 times to collect as much as colonic crypts. Next, these crypts were spun down at 300 x g for 5 mins at 4˚ C. The crypts were resuspended in ice cold basal culture media. 10 μl of suspension was taken in small petri dish and counted for crypts under light microscope. After counting, approximate 200 colonic crypts were seeded in 40 μl of Matrigel per well and 50% WNR-conditioned media (WNR cell line L-WRN (ATCC CRL3276) in 24 well plate at 37˚C in a 5% $CO_2$ environment. Importantly, 50% WNR media was supplemented with 10 μM ROCK inhibitor (Y-27632) for first 2 days that improve the survival of crypts derived colon organoids. Normal 50% WNR media was added to all the wells after 2 days of incubation. After 7 days of incubation, colon organoids were formed from colonic crypts and matrigel was disrupted using 1 ml pipette and then split into more wells. Next, every 4 to 5 days, colon organoids were passaged and seeded for MUC2 immunofluoroscence imaging in IBIDI slides and RNA isolation in 24 well plate.

## Bright field imaging and quantification of intestinal organoids

Bright field images of intestinal organoids were captured using a EVOS2 microscope with $CO_2$, temperature, and a humidity-controlled incubation chamber (Thermo-Fisher Scientific). Z-stacks covering the entire Matrigel droplet with a 2X objective was captured for quantification. 2D morphological properties of organoid objects as well as their classification were gathered using a custom analysis program written in python based on opencv2. Further details of the program can be found in this pre-print [8]. To find the outline of organoids, the brightness of images was autoscaled to max brightness, and a canny edge detection algorithm was used on each individual z-plane. Pixel groups below a certain size were removed and a minimal projection of the edges was generated. This image was used to define the contour of objects.

Opencv2s watershed algorithm was used to split somewhat overlapping objects from each other with the center of the object defined as the pixel furthest from the edge of the object. Each defined object contour was used to extract a 120x120 pixel picture of the object on a white background from a minimal projection of the original z-stack. These images were used to classify the organoid as either "Junk", "Budding" or "Spheroid" with a convolutional neural network implemented using Tensorflow and Keras [46].The network was trained on about 25 000 manually classified 120x120 images prepared as just described. Images were reviewed to see that segmentation and classification was adequate. [8]. No manual correction was done for the image quantification, but quantification was viewed for every image to see that segmentation and classification was adequate.

### Organoid cultures and IF staining

For immunofluorescence assays, organoids were seeded in pre-warmed 8 well chamber IBIDI slides. After incubation, organoids were fixed in 4% paraformaldehyde with 2% sucrose for 30 minutes. Wells with fixed organoids were washed twice with sterile PBS for 5 mins each. Next, organoids were permeabilized with 0.2% Triton X 100 prepared in PBS and incubated for 30 min at room temperature. To block free aldehydes groups and prevent high background signal, organoids were incubated with 100 mM glycine for 1 h at room temperature. The organoids were then incubated in the blocking buffer (1% BSA, 2% NGS diluted in 0.2% Triton X 100 in PBS) in a humidified chamber for 1 h at room temperature. Organoids were incubated in the diluted primary antibody in the same blocking buffer for overnight at 4˚C. Next day, the primary antibody anti ki67 (1:200 abcam ab15580), MUC2 (1:200 Santa Cruz Biotechnology sc-15334), β- catenin (1:200 Santa Cruz Biotechnology sc-7963), Filamin A (1:200 Invitrogen PA5-82043), RELMβ (1:200, Peprotech 500-p215) solution was decanted, and wells were washed three times for 10 minutes each with PBS with slight agitation. Organoids were incubated with secondary antibody (1:500, Goat anti-Rabbit IgG Alexa Fluor 488 Invitrogen A-11034 and Goat anti-Mouse IgG Alexa Fluor 647 Invitrogen A-21236), Ulex europaeus agglutinin-1 (UEA1, 1:500 Vector Laboratories RL-1062) and DAPI (1:1000) counterstained for overnight at 4˚C in the dark. Next, secondary antibody solution was decanted, and wells were washed three times with PBS for 10 minutes in agitation. After staining, wells were mounted in 250 μL of Fluoromount G and were visualized with a confocal microscope.

### Quantification of fluorescence intensity and specific cell markers

For quantification of fluorescence intensity in organoids, confocal images were opened using FIJI software. Images were stacked using the Z project and sum slices were made. After that, all the color channels were split and threshold for each channel was calculated. Next, organoids were selected manually and fluorescence intensity for each color channel was calculated per each organoid area. For normalization, the fluorescence intensity of target protein such as Ki67, Filamin A and MUC2 was divided by fluorescence intensity of DAPI and plotted in the graphs. Graphs show quantification of 2 to 3 biological replicates, with 100–150 small intestine organoids and colon organoids (≈50) per condition and each dot represents one organoid. For quantification of RELMβ[+], MUC2[+], Ki67[+]and UEA1[+]cells in organoids and crypts, these positive cells were manually counted in ZEN 2.3 blue edition software.

### FITC dextran assay to assess epithelial barrier integrity

FITC dextran assay was performed in naive, WT and cKO mice infected with *C. rodentium* for 6 days. Briefly, on the day 6 post infection, 4 kDa fluorescein isothiocyanate (FITC)-dextran (Sigma-Aldrich, FD4-1G) was dissolved in PBS at a concentration of 80 mg per ml. After that,

mice were orally gavage with 150 µl of FITC-dextran and food was withdrawn for next 4 hours. After 4 hours, mice were anesthetized using isoflurane and blood was collected through cardiac puncture. The blood was added immediately to the tubes containing heparin with final concentration of 1 IU. The tubes with blood were mixed properly and kept on ice. To collect serum, blood samples were centrifuged at 1,000 x g for 12 minutes at 4°C. Serum was collected and diluted to 1:1 in PBS. After that, 100µl of diluted serum was added to the 96 well black-walled, clear-bottom plates in replicates. In addition, a standard curve was prepared by dilution 80 mg per ml FITC-dextran to 25 µg, 12.5 and until 8 half dilutions. The fluorescence of FITC- dextran was acquired at fluorometer (POLARstar Omega) at an excitation wavelength of 485 nm and 520 nm emission wavelength. After acquisition, a standard curve was plotted to determine the concentration of FITC-dextran in serum derived from different mice groups.

## Western blot analysis

Total protein was extracted from the organoids. Briefly, organoids were harvested and resuspended in RIPA Lysis and Extraction Buffer supplemented with PhosSTOP Phosphatase Inhibitor Cocktail for 10mins in ice. Proteins (50µg) were separated by NuPAGE Bis-Tris protein precast polyacrylamide gels and transferred to nitrocellulose membranes using iBlot 2 Gel Transfer device. The membranes were incubated with 5% blocking buffer (5% BSA in TBST) for 1h at room temperature. The membranes were further incubated with the primary antibodies against p-STAT3 (1:2000, Cell Signaling Technology, 9145), STAT3 (1:1000, Cell Signaling Technology, 9139) and GAPDH (1:5000, abcam ab125247) for overnight at 4°C. Next day, after three washes with TBST for 5 mins, membranes were incubated with secondary antibody conjugated with horseradish peroxidase for 1 h at room temperature. Again, membranes were washed three times with TBST for 5 minutes and were visualized with Super Signal West Femto Maximum Sensitivity Substrate and images were acquired using Licor Odyssey detection system.

## Mesenteric lymph nodes isolation and CD3/CD28 restimulation

The mesenteric lymph nodes were isolated from naïve or infected mice and kept in sterile PBS. Harvested lymph nodes from individual mouse were passed through 70 µm cell strainer for single cell suspension. The cell viability was assessed using Trypan blue and counted using Countess automated cell counter (Invitrogen). The cells were resuspended in DMEM and adjusted to $2 \times 10^5$ cells per well. A solution of anti-mouse CD3e (100314, Biolegend) was prepared at 5 µg/mL in sterile PBS and 50 µl of antibody solution was added into 96 well plate followed by incubation at 37°C for 2 hours. The antibody solution was removed with a multichannel pipettor and single cell suspension of lymph node cells ($2 \times 10^5$ cells per well) was added to respective wells in coated 96 well plates. Next, soluble anti-mouse CD28 (102112, Biolegend) was added to cells at 2 µg/mL. Cells were incubated for 4 days to induce proliferation of cells. After incubation, cells were centrifuged and supernatant was collected for cytokine analysis by ELISA.

## Flow cytometry

Colon from WT and cKO mice were dissected and flushed by PBS. Colon was open longitudinally and cut into small piece of fragments. These fragments were further washed with cold PBS until the supernatant was clear. Next, the colon fragments were incubated with crypts isolation buffer (20mM EDTA in PBS) for 45 minutes at cold room with slight agitation. After incubation, EDTA buffer was removed and fragments were vigorously resuspended in 20ml of ice chilled PBS by shaking and 10mL pipette. The supernatant with crypts was collected in

50ml falcon tube coated with FBS. Next, supernatant with crypts was centrifuged at 300 g for 5 minutes at 4˚C to pellet down the crypts. Crypts were further resuspended in TryplE express for 30 minutes at 37˚C and were dissociated with pipette every 5 minutes. Digested crypts cells were incubated with anti-mouse TruStain Fc (1:400, BioLegend 101320) for 5 minutes in ice. Cells were further stained with Alexa Fluor 488 anti-mouse SCA1 antibody (1:200, BioLegend 108116) and other epithelial cells markers (anti-mouse CD326 Brilliant Violet 605, 1:200 Biolegend 118227, and PerCP/Cy5.5 anti-mouse CD24, 1:200 Biolegend 101824) for 20 minutes at 4˚C, twice washed with PBS and incubated with DAPI for 3 minutes in the last step. Before acquisition, cells suspension was passed through a 70 μm filter mesh. All the samples were acquired using a BD LSR II flow cytometer (BD Biosciences) and analyzed using FlowJo software.

## Statistical analysis

The statistical analysis was performed using the Graph Pad Prism. Unpaired two-tailed Student's t test and One-way analysis of variance with Tukey's Multiple Comparison Test were used to determine the Statistically significant differences between the different experimental groups and P value of $<0.05$ was considered significant. All the Statistical tests used are described in the figure legends. All numerical data used to make the figures is reported in S1 Data.

## Supporting information

**S1 Fig. Incomplete deletion of *Lsd1* in *Villin*-Cre *Lsd1*^f/f (cKO) mice.** (A) Confocal microscope images of whole colon swiss roles stained for LSD1 (green) expression comparing naive WT and cKO mice. DAPI (blue) was used as a nuclear counterstain. (B) LSD1 expression in WT and cKO colon WT(n = 5) and cKO (n = 3).
(TIF)

**S2 Fig. Gating strategy for SCA-1 in colonic crypts.** (A) Gating strategy used for flow cytometry plots to determine DAPI- (live) EPCAM+ (epithelial) cells.
(TIF)

**S3 Fig. Escaper crypts in cKO mice have WT features.** (A) Immunofluorescence staining to check co-localization of LSD1 (green) and RELMβ (green) in colon sections of cKO mice infected with *C.rodentium* for 6 days. DAPI (blue) was used as a counterstain. Original magnification: 10 X and scale bar- 50μm. (B) Immunofluorescence staining of FLNA (green) in colon sections from naïve cKO mice. White arrow indicates that there is no FLNA expression in WT escaper crypt. Original magnification: 20 X and scale bar- 50μm.
(TIF)

**S4 Fig. GSK-LSD1 treatment downregulates Goblet cells in organoids.** (A) RT- qPCR for *Muc2* expression in organoids left untreated or treated with GSK-LSD1 for 4 days. Expression was normalized to housekeeping gene *Hprt*. (B) Immunofluorescent images of MUC2 (green), UEA1 (red) and DAPI (Blue) in WT organoids that were cultured in normal culture medium (ENR = EGF, NOGGIN, R-SPONDIN) or in ENR medium supplemented with GSK-LSD1 for 4 days. Scale bar is 50 μm.
(TIF)

**S5 Fig. cKO colon organoids have negligible LSD1 and *Muc2* expression.** (A) Immunofluorescent staining of LSD1 (green), UEA1 (red) and DAPI (blue) in WT and cKO colon organoids was performed. Scale bar is 50 μm. (B) RT- qPCR for *Muc2* expression in WT and cKO

colon organoids cultured in ENR for 4 days. Expression was normalized to housekeeping gene *Hprt*.
(TIF)

**S1 Table. Supplementary table 1 for primers information.**
(DOCX)

**S1 Data. Numerical values: Data file that includes all numerical values used to make graphs.**
(XLSX)

## Acknowledgments

We thank Drs. Stuart Orkin and Sylvie Robine for kindly sharing mouse strains. We thank the imaging (CMIC), genomics (GCF), and animal care (CoMed) core facilities that assisted in this work (NTNU). BAV is the CH.I.L.D. Foundation Chair in Pediatric Gastroenterology.

## Author Contributions

**Conceptualization:** Naveen Parmar, Håvard T. Lindholm, Menno J. Oudhoff.

**Data curation:** Naveen Parmar, Kyle Burrows, Pia M. Vornewald, Håvard T. Lindholm, Rosalie T. Zwiggelaar, Mara Martín-Alonso, Madeleine Fosslie, Colby Zaph, Menno J. Oudhoff.

**Formal analysis:** Naveen Parmar, Kyle Burrows, Pia M. Vornewald, Håvard T. Lindholm, Rosalie T. Zwiggelaar, Mara Martín-Alonso, Madeleine Fosslie, John Arne Dahl, Menno J. Oudhoff.

**Funding acquisition:** John Arne Dahl, Colby Zaph, Menno J. Oudhoff.

**Investigation:** Naveen Parmar, Kyle Burrows, Pia M. Vornewald, Håvard T. Lindholm, Rosalie T. Zwiggelaar, Alberto Díez-Sánchez, Mara Martín-Alonso, Madeleine Fosslie, John Arne Dahl.

**Methodology:** Kyle Burrows, Pia M. Vornewald, Rosalie T. Zwiggelaar, Alberto Díez-Sánchez, Mara Martín-Alonso, Madeleine Fosslie.

**Project administration:** Naveen Parmar.

**Resources:** Bruce A. Vallance.

**Supervision:** John Arne Dahl, Colby Zaph, Menno J. Oudhoff.

**Validation:** Alberto Díez-Sánchez, Bruce A. Vallance.

**Visualization:** Naveen Parmar, Håvard T. Lindholm.

**Writing – original draft:** Naveen Parmar, Menno J. Oudhoff.

**Writing – review & editing:** Naveen Parmar, Bruce A. Vallance, John Arne Dahl, Colby Zaph.

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
