## [Decision Letter · Decision Letter 0]

1 Feb 2021

Dear Mr. Oudhoff,

Thank you very much for submitting your manuscript "Intestinal-epithelial LSD1 controls goblet cell differentiation and effector responses required for gut immunity to bacterial and helminth infection" for consideration at PLOS Pathogens. As with all papers reviewed by the journal, your manuscript was reviewed by members of the editorial board and by several independent reviewers. In light of the reviews (below this email), we would like to invite the resubmission of a significantly-revised version that takes into account the reviewers' comments.

The revised version of your manuscript has been reviewed by 3 of the 4 reviewers who initially had the first version. While all of them recognized the clear improvement of the manuscript, one of the reviewers (an expert in intestinal epithelial homeostasis) found that there are still some experiments that are necessary to justify your conclusions. I agree with him that it is necessary for you to meet his expectations (and in particular the use of colon organoids) while being more critical of the results and the conclusions that follow.

We cannot make any decision about publication until we have seen the revised manuscript and your response to the reviewers' comments. Your revised manuscript is also likely to be sent to reviewers for further evaluation.

Sincerely,

Eric Oswald, Ph.D., D.V.M.

Associate Editor

PLOS Pathogens

Nina Salama

Section Editor

PLOS Pathogens

Kasturi Haldar

Editor-in-Chief

PLOS Pathogens

orcid.org/0000-0001-5065-158X

Michael Malim

Editor-in-Chief

PLOS Pathogens

orcid.org/0000-0002-7699-2064

The revised version of your manuscript has been reviewed by 3 of the 4 reviewers who initially had the first version. While all of them recognized the clear improvement of the manuscript, one of the reviewers (an expert in intestinal epithelial homeostasis) found that there are still some experiments that are necessary to justify your conclusions. I agree with him that it is necessary for you to meet his expectations (and in particular the use of colon organoids) while being more critical of the results and the conclusions that follow.

Reviewer's Responses to Questions

**Part I - Summary**

Reviewer #1: The authors have now refocused their story on the role of LSD1 in goblet cell differentiation and activity using two different models of infection.

In its present form the manuscript is way more convincing and vehicle a clear message.

Reviewer #2: This revised version of the manuscript has been greatly improved. The authors have done a nice job by addressing most of the comments and adding supporting additional data.

Reviewer #3: This study explores the role of the histone demethylase LSD1 in intestinal pathophysiology, which is a topic of broad interest to the medical, as well as cell biology community. In summary, the authors characterised the effects of LSD1 invalidation on colonic epithelial homeostasis and repair upon infection, using an in vivo approach (colon tissue). Their work points to an important role of LSD1 in epithelial tissue homeostasis via the regulation of a goblet cell-specific response to infection.

The authors claim that LSD1 also controls goblet cell differentiation; however, this cannot be assessed solely on the basis of markers and approaches used in this study.

I also have reservations regarding (i) the use of small intestinal organoids to model a colon-specific response to infection, and (ii) the fact that the published RNAseq data the authors rely on is from small intestinal crypts; this confounds their GSEA and GO analyses in the current paper. The strongest conclusions in this paper are obtained from experiments on colon tissue, which is indeed the relevant tissue for this study, considering the pathophysiological scope of this work using infection models that specifically target the large intestine. I find that more data would be needed to reinforce certain conclusions, as well as a more thorough discussion of the observed effects, especially with regard to the consequences of upregulation of filamin A and/or other cytoskeletal regulators/organisation, and how this could reflect on goblet cell-specific response, observed in their experiments.

**Part II – Major Issues: Key Experiments Required for Acceptance**

Reviewer #1: N/A

Reviewer #2: None

Reviewer #3: 1. Assessing goblet cell differentiation based only on immunostaining for Muc2 and RELMB, PAS staining or lectin-binding capacity (UEA1), is problematic. As the authors themselves stated in their response to reviewer 3, (2/Figure 6) “MUC2 can be secreted, and thus these cells cannot be counted; UEA1, in our hands, is not always a reliable marker for goblet cells, especially in organoids (i.e. we have MUC2+ cells that are UEA1-)”. Also, the relationship between Muc2 and RELMB does not seem to be trivial (RELMB was dramatically upregulated in Muc2-/- mice; see Morambudi et al. 2016; PMID: 26813339). Moreover, judging from the image of the colon crypts (63X; E-cad, DAPI) that was provided in response to my concerns in the previous revision, many cells with goblet-like morphology are still present in the cKO. Thus, considering the complexity of assessing cell identity/differentiation, my suggestion is that the focus of the study should remain on goblet cell effector responses in LSD1-KO and infection models, which are well established in this study, and not in a role of LSD1 in goblet differentiation, for which the evidence is not enough.

2. Authors should address the role of LSD1-KO in colon organoids, in order to be consistent with the organ and epithelial tissue they are addressing in their in vivo experiments. Authors’ reasoning to use small intestinal (SI) organoids as a model in this study was based on less efficient recombination of LSD1-KO in the colon and, more importantly, the fact that colon organoids culture requires Wnt, which results in incomplete differentiation. According to Sato et al. 2011a (PMID: 21889923), Wnt can be removed (“differentiation medium”) which then allows for complete differentiation. Please note that Sato et al. assessed epithelial cell differentiation in cultured colon organoids by measuring mRNA expression of various markers (Muc2, ChgA, Alpi..) and thus did not solely rely on immunostaining or lectin binding.

3. Authors claim that “FILAMIN A, a key protein in mechanosensing and mechanotransduction (references are missing), is a direct target of LSD1 demethylase activity” yet they don’t try to functionally validate this proposed role of filamin A in the colon, so this seems to be an exaggeration. I suggest that the authors refrain from such statements unless they are able to tackle and explain the role of filamin A in this context, assuming there were more candidates coming from the screen. As it stands, filamin A and its increased expression in this study represents an example confirming their RNAseq screen and the follow-up analysis.

4. Authors used ROCK1 inhibitor to perturb “cytoskeletal organization”, which in itself sounds very vague. They should explain the rationale for this approach and discuss the results of this treatment in the context of what is known or proposed for this kinase inhibitor, including the maintenance and differentiation of stem cells, in addition to the well-studied effect on actomyosin contractility, which was not mentioned or discussed either.

**Part III – Minor Issues: Editorial and Data Presentation Modifications**

Reviewer #1: (No Response)

Reviewer #2: None

Reviewer #3: 1. Line 452: Methods: “Bright field imaging and quantification of intestinal organoids” - instead of referring to their preprint, authors should explain in this paragraph how their images were quantified.

2. Line 100-101: LSD1 expression in colon cKO is not quantitated.

3. Fig. 1G, H, S2B: Western blot should be repeated and quantified (gel band densitometry). Authors need to clearly assess whether STAT phosphorylation response is the same or weaker upon IL-22 induction in cKO. The confocal images of organoids in S2B were not quantified either.

4. Line 131-132: increased mortality and weight loss are relatively mild, especially at day 6 pi. Why not looking at later stages?

5. Line 137-139: the effects of Citrobacter infection on the gut barrier permeability should be more clearly explained in the text. The barrier permeability was increased upon Citro infection, but there was no significant difference between the ctrl and the KO at day 6 pi.

6. Figs. 2-3C, F, G: Authors should provide data also for naïve crypts (ideally for both wt and cKO).

7. Fig. 3D, F: Authors should be explicit on whether they analysed crypts from the same region of the colon, as crypt length varies significantly between different regions of colon (PMID: 24205248)

8. Fig. S3A: These images are seemingly not from the same focal plane (or perhaps not the same region), so it is impossible to determine “colocalization”; also, “escaper” crypts should be somehow highlighted.

9. Line 182-183: can authors explain why the crypt length is different for WT and cKO between Citro and T. muris conditions (Fig. 3C vs 4G)?

10. Line 197: “we found that goblet cell differentiation and maturation is much more

impaired in the colons of naïve cKO mice, as shown by reduced numbers (differentiation)…”. Authors should be more careful – they did not show numbers of goblet cells (they show number of Muc2+ and UEA1+ cells per crypt, as stated in Fig. 5B and C).

11. Line 224: Authors used data from RNAseq performed on SI crypts cKO/WT and GSK-LSD1/untreated organoids and gene enrichment analysis, instead of performing this analysis on colon tissue/organoids. It should be explained why they did not analyse colon tissue/organoids.

12. Line 259: hyphen missing (LSD1 controlled)

13. Line 263: "induce MUC2 expression" would be more correct

14. Line 295: remove colitis, as those experiments were removed

15. Line 316: GFI1-KO mice were shown to have decreased goblet cell numbers/maturation, and not a complete lack of – this should be corrected.

PLOS authors have the option to publish the peer review history of their article (what does this mean?). If published, this will include your full peer review and any attached files.

Reviewer #1: **Yes: **tiffany bouchery

Reviewer #2: **Yes: **Chamaillard

Reviewer #3: No
---

## [Editor Report · Decision Letter 1]

14 Mar 2021

Dear Mr. Oudhoff,

We are pleased to inform you that your manuscript 'Intestinal-epithelial LSD1 controls goblet cell maturation and effector responses required for gut immunity to bacterial and helminth infection' has been provisionally accepted for publication in PLOS Pathogens.

Best regards,

Eric Oswald, Ph.D., D.V.M.

Associate Editor

PLOS Pathogens

Nina Salama

Section Editor

PLOS Pathogens

Kasturi Haldar

Editor-in-Chief

PLOS Pathogens

orcid.org/0000-0001-5065-158X

Michael Malim

Editor-in-Chief

PLOS Pathogens

orcid.org/0000-0002-7699-2064
---

## [Editor Report · Acceptance letter]

25 Mar 2021

Dear Mr. Oudhoff,

We are delighted to inform you that your manuscript, "Intestinal-epithelial LSD1 controls goblet cell maturation and effector responses required for gut immunity to bacterial and helminth infection," has been formally accepted for publication in PLOS Pathogens.

Best regards,

Kasturi Haldar

Editor-in-Chief

PLOS Pathogens

orcid.org/0000-0001-5065-158X

Michael Malim

Editor-in-Chief

PLOS Pathogens

orcid.org/0000-0002-7699-2064